# AN AUTOMATIC OPERATION BATCHING STRATEGY FOR THE BACKWARD PROPAGATION OF NEURAL NETWORKS HAVING DYNAMIC COMPUTATION GRAPHS

## ABSTRACT

Organizing the same operations in the computation graph of a neural network into batches is one of the important methods to improve the speed of training deep learning models and applications since it helps to execute operations with the same type in parallel and to make full use of the available hardware resources. This batching task is usually done by the developers manually and it becomes more difficult when the neural networks have dynamic computation graphs because of the input data with varying structures or the dynamic flow control. Several automatic batching strategies were proposed and integrated into some deep learning toolkits so that the programmers don't have to be responsible for this task. These strategies, however, will miss some important opportunities to group the operations in the backward propagation of training neural networks. In this paper, we proposed a strategy which provides more efficient automatic batching and brings benefits to the memory access in the backward propagation. We also test our strategy on a variety of benchmarks with dynamic computation graphs. The result shows that it really brings further improvements in the training speed when our strategy is working with the existing automatic strategies.

## 1 INTRODUCTION

Recent years, neural networks (NN) have been applied and shown their great effects on a lot of Natural Language Processing (NLP) topics like Sentiment Classification(Socher et al., 2013; Tai et al., 2015), Named Entity Recognition (NER)(Lample et al., 2016), Machine Translation (MT) (Sutskever et al., 2014; Wu et al., 2016; Eriguchi et al., 2016), Question Answering (QA) (Kumar et al., 2016) and so on. It has also become crucial to improving the speed of training neural networks for the research and development of deep learning models and applications since the neural networks as well as the training sample datasets become larger and larger, which leads to days of training time. Taking *Machine Translation* as an example topic, Table. 1 summarized both the accuracy performance and computing performance of each state-of-the-art work on WMT 2014 English-to-French Machine Translation task since 2014. We can see that the training phase of machine translation is time-consuming and it takes at least days to finish even accelerated by multiple GPUs. Therefore, it's becoming more and more important to take the computing performance of training neural networks into consideration.

In order to maximize the use of computing resources provided by modern parallel processors such as multi-core CPUs and many-core GPUs, *batching* is widely used in the existing implementations of neural networks for deep learning. This is because the operations batched together can be evaluated simultaneously on modern CPUs and GPUs and the evaluation is much faster than evaluating the operations one by one. For example, programmers can execute a single matrix-matrix multiplication using an optimized BLAS call instead of multiplying tens or hundreds of different vectors by the same matrix one by one and the former implementation is much faster on both CPUs and GPUs.

Usually, it's the programmers who implement the neural networks in charge of crafting efficient batching by hand. This is easy in some cases where the computation graphs of neural networks are stable and the inputs and outputs are represented as fixed sized tensors (e.g., training LeNet-5 (LeCun et al., 1998) with MNIST dataset in which all the images have the same fixed size). In

Table 1: Accuracy and Performance of State-of-the-art works for NMT on WMT2014 English to French Dataset.

| Paper | Model | BLEU | Training Time |
|---|---|---|---|
| Cho et al. (2014) | Phrase table with neural features | 34.50 | 3 Days |
| Sutskever et al. (2014) | Reranking phrase-based SMT best list + LSTM seq2seq | 36.5 | 10 Days with 8 GPUs |
| Wu et al. (2016) | Residule LSTM seq2seq + RL refining | 41.16 | 6 Days with 96GPUs |
| Gehring et al. (2017) | seq2seq with CNN | 41.29 | 37 Days with 8 GPUs |
| Vaswani et al. (2017) | Attention mechanism | 41.0 | 3.5 Days with 8 GPUs |

these cases, the programmers write code for the computation of processing an individual training sample and add another leading dimension to the tensors representing the inputs and outputs so that multiple tensors can be packed into and represented as only one. Then the program will process the new tensor which actually contains the data from multiple inputs as an individual one using the high performance computing technology like SIMD instructions or BLAS library to execute it efficiently in parallel (the correctness is guaranteed by the batch learning which allows multiple inputs to be processed using the same parameters).

However, the batching tasks tend to become difficult for programmers to finish it manually when the inputs have different structures or the computation graphs of the neural networks is unstable during the training phase. Neural networks having dynamic computation graphs are common in models or application for NLP. For example, the inputs of neural networks based NLP model are always sentences represented as word sequences with different lengths or parse trees with different structure (Socher et al., 2011), and the corresponding unfolded computation graphs of recurrent or recursive neural networks (Elman, 1990) vary when processing inputs with different structures. In some cases where the inputs are sequences with different lengths, programmers can pad the shorter ones with the same special symbols so that all the sequences have the same lengths. This approach, which is called *padding*, is widely used in existing neural network based NLP models and helps with utilizing batching in the execution. However, the strategy padding strategy still has limitations: the computation on the padded symbol is considerable but totally a waste (Qiao et al., 2018) and it is also difficult to apply this strategy on neural networks with more complex architectures, like Tree-LSTM (Tai et al., 2015). Indeed, many implementations of neural networks having dynamic computation graphs choose to avoid batching data and operations entirely or just use single instance training.

In order to apply more efficient batching without programmer's manual work, researchers tried to develop automatic batching strategies that can instruct the deep learning frameworks or libraries to apply the operation batching by itself. The programmer should only write the code that defines the computation of processing one individual instance and leave the details of operation batching to the frameworks or libraries. Google proposed and implemented their automatic operation batching strategy in TensorFlow Fold (Looks et al., 2017). Another strategy called *Agenda-based batching* was also proposed in Neubig et al. (2017b) and implemented using DyNet toolkit (Neubig et al., 2017a). Both of these operation auto-batching strategies bring huge gains in computing performance of neural networks having dynamic computation graphs.

Unfortunately, there are still limitations in both of these strategies. In the training phase, the auto-batching strategy generates an execution order of the operations in the computation graphs and this execution order determines which operations can be batched and executed on the same step. Usually, the execution order is generated according to the dependency between the operations in the forward propagation of computation graph. In the backward propagation, the framework just executes the operations (calculating the derivative of each node) using a reverse of that generated execution order. This scheme leads to missing some batching opportunities in the backward propagation. For example, there are some operations with the same type share the same parameters but can't be batched in the forward propagation. But the calculations of the derivatives from the results of those operations to the shared parameters can be executed together. If we are able to utilize this kind of operation batching opportunity in the backward propagation, we can achieve further improvement of the training speed of neural networks having dynamic computation graphs.

In this paper, we propose our automatic operation batching strategy for the backward propagation of neural networks having dynamic computation graphs. This strategy explores and utilizes the operation batching opportunities described as above in the backward propagation. It also helps to execute those new batched operations with efficient memory accesses. The combination of our strategy and the existing ones can provide further improvement in the speed of the training phase.

## 2 BACKGROUND

### 2.1 OPERATION BATCHING

Operation batching is the most common way to enable parallelism in deep learning. It groups the same operations from multiple training samples together and executes them simultaneously. Operations from the same training can also be batched if they have the same type and no dependency on each other. Since modern parallel processors (CPUs and GPUs) have powerful SIMD processing instructions, operation batching often brings large gains in the computation efficiency.

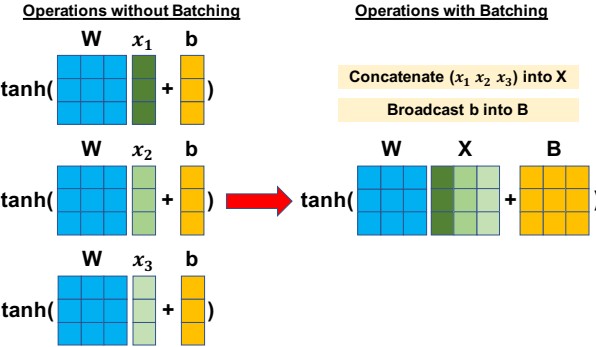

Figure 1: Operation batching for an affine transformation followed by a tanh element-wise activation function

Figure 1 illustrates the basic concept of operation batching. The function $y = tanh(Wx + b)$ is called three times for the independent input vector $x_1, x_2, x_3$. For efficient execution, we can concatenate the three vectors into a matrix $X$ and broadcast the bias vector $b$ into a matrix $B$, then execute the affine transformation as $Y = tanh(WX + B)$. The result $Y$ consists of three column vectors $y_1, y_2, y_3$, which are the corresponding results of the function calls for $x_1, x_2, x_3$. In this case, the operation batching brings benefits for computation efficiency from three aspects: First, the matrix $W$ is loaded from the lower level memory to the upper-level memory for only once, which results in a better locality to the memory access behavior; Second, the element-wise activation function *tanh* is called only once, which asks for fewer times of computation kernel launch; Third, the matrix-matrix product between $W$ and $X$ can be implemented using an optimized BLAS library, which makes efficient use of hardware resources.

### 2.2 MANUAL OPERATION BATCHING

It is not easy to apply the operation batching on the neural networks having dynamic computation graphs. Figure 2 illustrates the batch learning using a basic vanilla Recurrent Neural Network (RNN) which takes a sequence of word embedding vectors as the input. The final hidden state of this RNN is used to make a prediction and calculate the training loss. In the example shown in the left part, a computation graph of mini-batch learning is shown. The batch size is 3 and each training instance has a different length. An ideal set of batched operations for this given example is ($Op_1^1$, $Op_1^2$, $Op_1^3$), ($Op_2^1$, $Op_2^2$, $Op_2^3$), ($Op_3^1$, $Op_3^3$), ($Op_4^1$), ($L^1$, $L^2$, $L^3$), where $Op_t^i$ represents the operations of RNN cell at the $t$ th step in the $i$ th instance. The operations in the same batch share the same operation type and the same size of inputs and outputs, meanwhile, they don't have dependencies on each other. After batching operations like that, we can execute them in an efficient way as described above. Nevertheless, it's really difficult to implement an ideal batching for every mini-batch since the length distributions of sequences vary in each mini-batch since we can't count on the programmers

to manually implement that with hard code for each one. Besides the operation batching, the multi-processing, which means distributing training instances from a mini-batch to multiple threads or processes then synchronizing after processing each mini-batch, is another way to enable parallel processing. The multi-processing is easy to implement but unfortunately, its execution is generally inefficient since the similar operations across the training instances are performed separately and sequentially.

The benefit of operation batching, which helps to process training instances in parallel and taking advantage of data-parallelism in the training phase, is still attractive. One most common way to make use of the operation batching for training RNN-based models is to pad the input sequences with dummy elements so that all the training instances from a mini-batch have the same length. The right part of Figure 2 illustrates the corresponding computation graph after padding the training instances. In the practical usage of deep learning libraries or frameworks, it is always required to pad all the instances of training dataset so that they have the same length. After the padding, the computation graph of each mini-batch has the same structure and can be treated as a static one. The programmers can write the code for manual operation batching once and repeatedly use it to apply operation batching on every mini-batch. In order to keep the correctness of execution, the intermediate results and resulting loss from the computation at the dummy elements are masked out using a mask matrix.

Even though the padding strategy makes it possible to apply operation batching for the input sequences with different lengths from a mini-batch, it still has three disadvantages: First, the programmers are still in charge of implementing padding and masking, which increases the possibility of inefficient implementation and presenting bugs; Second, the computation on the dummy elements is totally unnecessary leads to the waste of computing resource; Third, for more complex neural networks having dynamic computation graphs like Tree-LSTM, it's almost impossible to apply the padding strategy on each training instance.

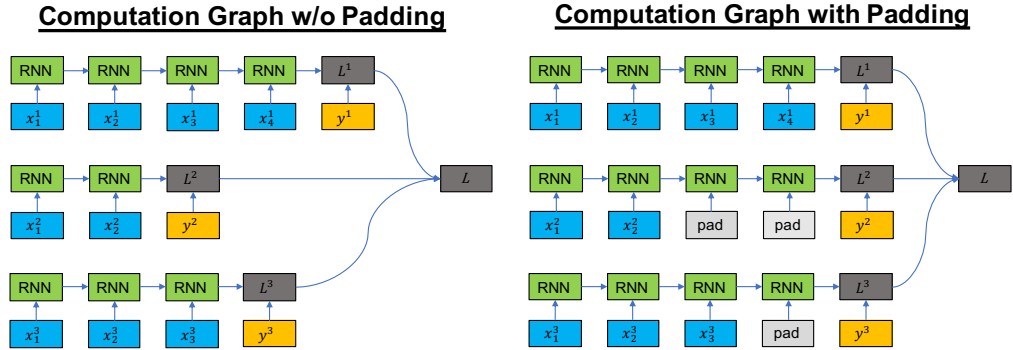

Figure 2: Two computation graphs of calculating the loss on a mini-batch consisting of three training instances for a neural network using vanilla RNN.

## 2.3 AUTOMATIC OPERATION BATCHING

Since the manual operation batching is still a pain point, researchers and engineers ask for the automatic operation batching, which means the runtime environment provided by the deep learning libraries or frameworks deal with the operation batching on-the-fly. In the computation graphs of neural networks, the operations are always represented as nodes while the dependencies among them are represented as directed edges. To apply the operation batching means analyzing a given computation graph and convert it into a new one. The nodes in the new graph represent the batches containing the operations that can be batched and executed together in the original one.

It usually takes two steps to apply an automatic operation batching strategy. The first step is to partition the operations in the original computation graph into compatibility groups, where the operations in the same group have the potential to be batched. The operations in the same group usually share the same type (element-wise operations such as *tanh* and *log*), even the same size of inputs/outputs and the same parameters (affine transformation such as *Wx+b*). The second step is to determine the

batch to which each operation belongs and the execution order of each batch in the new computation graph. Only the operations which come from the same compatibility group and have no dependency on each other can be assigned to the same batch. Meanwhile, the execution order should guarantee the dependencies among operations in the original computation graph. Two heuristic strategies have already been proposed to accomplish the second task, trying to find an optimal result that maximizes the amount of batched operations.

*Depth-based batching* (Looks et al., 2017) is implemented by assigning each node a depth of it in the original computation graph. The depth of a node is defined as the maximum length from a leaf node to itself. Nodes that have an identical depth and signature (which represents that they belong to the same compatibility group) are batched together. In this design, nodes have the same depth don't depend on each other and all the nodes will have a higher depth than its inputs. As a result, the operation batching can be done by the runtime environment automatically. However, this heuristic strategy has a shortcoming because it will miss good opportunities like batching the loss function calculations ($L^1$, $L^2$, $L^3$) in Figure 2 because the nodes represent these calculations don't have the same depth.

*Agenda-based batching* (Neubig et al., 2017b) is another automatic operation batching strategy that does not depend solely on depth. This strategy simulates the execution of the original computation graph and maintains an agenda that records all the available nodes which don't have unsolved dependencies. The simulation only traverses the nodes but do not perform the actual computation of operations. During the initialization, nodes have no coming inputs in the original computation graph are put into the agenda. At each iteration, nodes with the same signatures from the agenda are gathered into a single batch in the new graph and then removed from the agenda. Removing the nodes from the agenda means they have been "solved" in the simulation. If there is a successor of these batched and removed nodes having no unsolved dependency (coming input), the successor is put into the agenda. This process is repeated until all the nodes have been solved. The order in which the batches are generated in the simulation is used as the execution order of them in the new computation graph. During the simulation, there may be two groups of nodes that can be batched and removed existing in the agenda at one iteration. In order to prioritize the nodes in the agenda, there is a heuristic method based on the average depth of all nodes with their signatures. The nodes with a lower average depth will be batched and removed from the agenda earlier. With this heuristic method, the agenda-based batching strategy is able to overcome the depth-based one's shortcoming described above. As a result, the agenda-based batching strategy beats the depth-based one in the most cases of experiments conducted in the paper.

## 3 OPERATION BATCHING IN BACKWARD PROPAGATION

### 3.1 SHORTCOMINGS OF EXISTING STRATEGIES

Both of the two existing automatic operation strategies bring great improvement in the execution speed of neural networks having dynamic computation graphs, however, there is still a shortcoming in them which leads to missing good operation batching opportunities in the backward propagation of the training phase. Usually, the backward propagation is done by traversing the nodes in the reverse execution order of the forward propagation and calculating the derivative of loss with respect to the result of each node using the *chain rule*. At last, the derivatives of loss with respect to the parameters are calculated and used as gradients to adjust the weights. However, we should notice that the derivative of loss with respect to a node is the summation of derivatives calculated through different paths to node's successors. The calculation of different terms in the summation expression can be batched together.

In order to make the analysis of the shortcoming easy to understand, we use the example in Figure 2 for further explanation. Let's assume the RNN used in the example is the most simple vanilla one whose mathematical representation is as follow:

$$h_t = \sigma(W_{xh}x_t + W_{hh}h_{t-1} + b) \tag{1}$$

where $\sigma$ is an element-wise activation function. After applying the operation batching, the mathematical representation of the RNN part in the computation graph can be expressed as:

$$H_t = \sigma(W_{xh}X_t + W_{hh}H'_{t-1} + B) \tag{2}$$

where $X_t$ is the concatenated matrix of $x_t^1$, $x_t^2$, $x_t^3$ , $H_t$ is the concatenated matrix of $h_t^1$, $h_t^2$, $h_t^3$ (when $t$ is 3 or 4, there are few columns in the concatenated matrix) and $B$ is the broadcast matrix of $b$. $H_{t-1}'$ is a subset of $H_{t-1}$. When the number of batched operation at $t$ is less than that at $t-1$, $H_{t-1}'$ only collects the columns related to the calculation of $H_t$ from $H_{t-1}$.

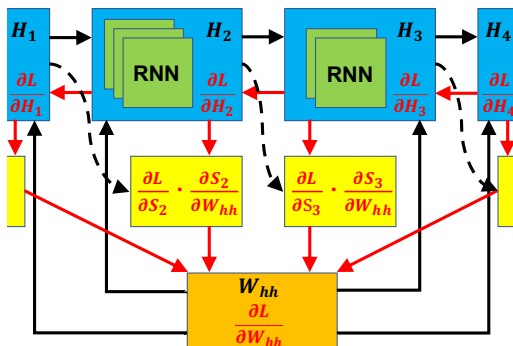

Figure 3: Part of the generated computation graph of RNN after applying operation batching. Each blue rectangle represents the batched operations of an RNN cell while the orange one represents the parameter matrix $W_{hh}$. Each yellow rectangle represents a term of the summation that calculates the derivative of loss w.r.t $W_{hh}$. In each rectangle, the variables with black color represent the parameters and the intermediate results calculated in the forward propagation and those with red color represent the derivatives calculated in the backward propagation. The black solid arrows represent the dependencies in the forward propagation while the red ones represent those in the backward propagation. The arrows of dashes represent the dependencies in the backward propagation, too. Since they represent the calculation of derivatives' dependencies on the intermediate results in the forward propagation, the arrows are also in black.

Figure 3 shows a part of the new computation graph generated by the operation batching, which mainly includes the computation related to the parameter matrix $W_{hh}$ in both the forward and the backward propagation. With the illustration through this figure, we discuss on the operation batching opportunities when calculating the derivative of loss with respect to $W_{hh}$. In the backward propagation, the derivative of loss with respect to $W_{hh}$ is calculated as:

$$\frac{\partial L}{\partial W_{hh}} = \sum_t \frac{\partial L}{\partial H_t} \frac{\partial H_t}{\partial W_{hh}} = \sum_t \frac{\partial L}{\partial S_t} \frac{\partial S_t}{\partial W_{hh}} \tag{3}$$

where

$$S_t = W_{xh} X_t + W_{hh} H_{t-1}' + B \tag{4}$$

In Formula 3, $\frac{\partial S_t}{\partial W_{hh}} = {H_{t-1}'}^{\mathrm{T}}$ and $\frac{\partial L}{\partial S_t}$ is calculated through $\frac{\partial L}{\partial H_t}$ (the details of calculation is determined by the activation function $\sigma$). It is noticeable that the calculation of each term of the last summation shares the same operation type and output size as well as similar input size. Meanwhile, they don't depend on each other. Therefore, these calculations can be batched and executed together. This opportunity will be missed if the batching execution in the backward propagation only takes the reverse execution order in the forward propagation because the corresponding calculations to the different term in the forward propagation depend on each other. For example, the calculation of $S_t$ depends on $S_{t-1}$ through $H_{t-1}$.

Generally, there are operation batching opportunities like what is described above in the computation graph where there are operations with the same type depending on each other and sharing the same parameters or the same intermediate results. This feature is common especially in the neural networks with recursive or recurrent structure. If we can utilize these opportunities, we will further improve the computing performance of the training phase through accelerating the execution of backward propagation.

## 3.2 BATCHING CALCULATION OF DERIVATIVES

In order to make use of the operation batching opportunities in the calculation of derivatives for efficient computation, we can concatenate $\frac{\partial L}{\partial S_t}$ (column-wise) and $\frac{\partial S_t}{\partial W_{hh}}$ (row-wise) from each term of the summation at $t$ to form two concatenated matrices. We denote them as $\frac{\partial L}{\partial S}$ and $\frac{\partial S}{\partial W_{hh}}$ respectively, where[1]

$$\frac{\partial L}{\partial S} = \left[ \frac{\partial L}{\partial S_1}, \frac{\partial L}{\partial S_2} \cdots \frac{\partial L}{\partial S_T} \right] \tag{5}$$

and

$$\frac{\partial S}{\partial W_{hh}} = \left[ \frac{\partial S_1}{\partial W_{hh}}^{\mathrm{T}}, \frac{\partial S_2}{\partial W_{hh}}^{\mathrm{T}} \cdots \frac{\partial S_T}{\partial W_{hh}}^{\mathrm{T}} \right]^{\mathrm{T}} \tag{6}$$

As a result, the calculation of the derivative loss with respect to $W_{hh}$ is transformed into

$$\frac{\partial L}{\partial W_{hh}} = \frac{\partial L}{\partial S} \frac{\partial S}{\partial W_{hh}} \tag{7}$$

After the transformation, the summation of a series of matrix-matrix products is transformed into one product.

This transformation brings computation efficiency from two aspects: (1) Fewer computation kernels are launched for the matrix-matrix product and the overhead comes from launching kernels can be reduced. What's more, it takes almost the same time to compute a large matrix-matrix product compared to a small one on the accelerators like GPUs. As a result, it's faster to compute a large matrix-matrix product than to compute a series of small matrix-matrix products. (2) The number of times to get access to the memory where the data of the derivative is stored can be reduced. Usually, the summation is implemented in the form of $M+ = M'$, where $M$ represents the result of the derivative while $M'$ represents the intermediate result of the calculation of each term of the summation. In the new way to calculate the derivative, it gets access to the memory where it is stored only once.

## 3.3 IMPLEMENTATION IN PRACTICE

We propose a strategy that enables automatic operation batching for the calculation of derivatives in the backward propagation and implement it using the DyNet toolkit. Given a computation graph of a mini-batch, the strategy takes four steps to finish the training phase:

1. Apply automatic operation batching using by-agenda or by-depth strategy. Execute the forward propagation in the execution order generated by the strategy.

2. Enumerate all the nodes[2] in the new computation graph, record the nodes having successors that share the same operation type but belong to different batches in a list.

3. Traverse the nodes in the reserve execution order in the forward propagation. When visiting a node that is not in the list, propagate the derivative of loss w.r.t this node to its inputs by calculating the derivatives of loss w.r.t them[3]. If the node has inputs that appear in the list, don't propagate the loss to those inputs but others. When visiting a node in the list, calculate the derivative of loss w.r.t this node using the operation batching described above.

4. Update the parameters using the calculated derivatives.

With this implementation, our proposed strategy takes advantages of both an existing automatic operation batching strategy and the operation batching opportunities in the calculation of derivatives. The backward propagation of the training phase is accelerated. Unavoidably, there is overhead introduced through the second and the third step. It also asks for extra execution time to concatenate the small matrices into a larger one because data copy is necessary for ensuring contiguous memory. In practice, the overhead is smaller compared to the gains and therefore further improvement in the training speed is provided.

---

[1]$T$ is the range of $t$

[2]Or only the parameter nodes for a better trade-off between the computing performance improvement and the overhead, which we use in the practical implementation

[3]The calculated derivatives w.r.t the inputs here, however, are individual terms in the respective summations.

## 4    EXPERIMENTAL EVALUATION

In this section, we report the experimental evaluation to answer two questions: (1) Can our proposed strategy provide further improvements in the training speed of neural networks having dynamic computation graphs that have already applied automatic operation batching? If so, how much is the further improvement? (2) How much are the gain and the overhead of our proposed strategy?

We conducted all the experiments on a server with a dual Intel Xeon 2.2GHz E5-2699 v4 CPUs[4] and 2 Nvidia Tesla P100 GPUs on it. We also use MKL and CUBLAS to speed up computation. In practice, we just use one GPU to do the experiments. When running experiments on CPU with MKL, we use only 22 threads and bind them to 22 cores on the same CPU through *numactl* command so that we can eliminate the influence on computing performance from the Non-uniform Memory Access (NUMA) (Lameter, 2013). The operating system is Ubuntu 16.04.4.

### 4.1    TRAINING SPEED OF DIFFICULT-TO-BATCH BENCHMARKS

To answer the first question, we test the training speed of three benchmarks using neural networks having dynamic computation graphs. The experiments are based on implementations in the DyNet benchmark repository [5]. All the three benchmarks are extraordinarily difficult for programmers to apply operation batching manually. The information about the three benchmarks and the corresponding parameters setting are described below:

- *BiLSTM*: This is a benchmark that trains a tagger using a bi-directional LSTM to extract features from the input sentence, which are then passed through a multi-layer perceptron to predict the tag of the word. The model used in this benchmark is based on the one proposed by Huang et al. (2015) and is trained and tested on the WikiNER English Corpus (Nothman et al., 2013). In the experiments, the word embedding size is set to 128 while the LSTMs in either direction containing 256 hidden states. The size of multi-layer perceptron is set to 32.

- *BiLSTM w/char*: This benchmark is similar to the above one but has something different. In the first benchmark, words that have a frequency of at least five use a special embedding for that word while in this benchmark, the less frequent words use an embedding calculated by running a bi-directional LSTM over the characters in the word. This model can improve generalization by using the spelling of low-frequency words. In the experiments, the char embedding size is 64 and the word embedding size is still 128. The size of hidden states in LSTM is 256 and the size of multi-layer perceptron is set to 32. The datasets used are the same as those in the first benchmark.

- *Tree-LSTM*: This benchmark is a sentiment analyzer based on tree-structured LSTMs. Tree LSTMs are trained on the Stanford Sentiment Tree-bank regression task (Socher et al., 2013), which is provided in the benchmark repository. In the experiments, the word embedding size is 256 and the hidden state size is 256, too.

In the experiments, we measure the training speed of each benchmark on both CPU and GPU while applying automatic operation batching under 4 different circumstances: (1) using by-depth strategy only; (2) using both by-depth strategy and our proposed strategy; (3) using by-agenda strategy only; (4) using both by-agenda strategy and our proposed strategy. We perform each experiment 8 runs and report the average speed and the corresponding confidence interval. For all the experiments, we set the mini-batch size as 64. The results of the experiments are shown in Table 2 and 3, where *By-depth+* represents using both by-depth strategy and our proposed one. So it is with *By-agenda+*. The training speed is represented as how many sentences are processed per second.

From the tables, we can see our proposed strategy really brings further improvements in training speed in most cases, especially when the experiments are executed on GPU.

---

[4]There are 44 cores and 88 threads in total.
[5]https://github.com/neulab/dynet-benchmark

Table 2: Sentences per second processed in the training phase of different benchmarks on CPU.

| Task | By-depth | By-depth+ | By-agenda | By-agenda+ |
|------|----------|-----------|-----------|------------|
| **BiLSTM** | $298.83 \pm 1.87$ | $295.30 \pm 1.35$ | $333.98 \pm 2.09$ | $330.26 \pm 1.13$ |
| **BiLSTM w/char** | $59.16 \pm 0.23$ | $85.78 \pm 0.30$ | $74.99 \pm 0.36$ | $108.30 \pm 0.37$ |
| **Tree-LSTM** | $766.84 \pm 3.57$ | $741.83 \pm 4.84$ | $824.89 \pm 3.03$ | $789.56 \pm 3.32$ |

Table 3: Sentences per second processed in the training phase of different benchmarks on GPU.

| Task | By-depth | By-depth+ | By-agenda | By-agenda+ |
|------|----------|-----------|-----------|------------|
| **BiLSTM** | $590.25 \pm 2.29$ | $666.56 \pm 1.78$ | $728.55 \pm 2.61$ | $800.50 \pm 4.04$ |
| **BiLSTM w/char** | $136.84 \pm 0.95$ | $173.44 \pm 0.39$ | $121.23 \pm 0.54$ | $132.87 \pm 0.70$ |
| **Tree-LSTM** | $1279.88 \pm 4.74$ | $1362.87 \pm 6.98$ | $1294.47 \pm 3.65$ | $1364.85 \pm 6.74$ |

## 4.2 GAIN AND OVERHEAD IN OUR PROPOSED STRATEGY

In order to explain why our strategy can contribute to further improvement in the training speed, we do a profiling of the second benchmark (a.k.a. BiLSTM w/char). We execute this benchmark with automatic operation batching using the by-agenda strategy and let the program stop after processing 4,992 (Max number that is a multiple of 64, which is the batch size, and less than 5000) sentences. We use the same experiment settings as described above and run the benchmark on the GPU. We measure the execution time of different part of the training phase so that we can know where the computing performance gain comes from and where our strategy introduces overhead.

In Table 4, we report the elapsed time of several important parts of BiLSTM w/char using by-agenda strategy and our strategy. The meaning of denotations in the table are:

- *Apply By-agenda* represents the time used to apply automatic operation batching using the by-agenda strategy in the forward propagation.

- *Find Special Nodes* represents the procedure described in the second of our strategy. In this procedure, the special nodes having successors that share the same operation type but belong to different batches are recorded.

- *Calculate Derivatives* represents the main part of the backward propagation, in which to calculate the derivatives of each node in the computation graph using the chain rule.

- *Special Nodes* represents the time used for calculating the derivatives of the special nodes recorded through the efficient way we proposed, The calculation of *Special Nodes* actually consists of two steps. We need to copy the data into contiguous memory first, then apply the matrix-matrix product using CUBLAS call.

- *Common Nodes* represents the time used for calculating the derivatives of other non-special nodes.

In the original by-agenda strategy, the calculation of nodes' derivatives is not divided into two parts explicitly like that of our proposed strategy, so we can't measure the elapsed time of each category. In order to show the comparison, we use the elapsed time to calculate the derivatives of common nodes in our strategy as that in the original by-agenda strategy (We believe the elapsed time won't change much).

From the table we can see, the overhead of our strategy mainly comes from the time used to copy data into contiguous memory. Even though, we can still accelerate the time used for calculating the derivatives of those special nodes from 5.66 seconds to 1.67 seconds, which brings a 3.39x speedup for this part. We can also conclude that if a neural network has a computation graph containing more nodes that can be treated as the special ones discussed above, our strategy will bring more gain in the computing performance of its training phase by accelerating the backward propagation.

Table 4: The elapsed time (s) of different parts of BiLSTM w/char using by-agenda strategy and our strategy, running on GPU.

|  | By-agenda | By-agenda+ |
|---|---|---|
| **Forward Propagation** | 14.55 | 14.72 |
| **Apply By-agenda** | 1.79 | 1.95 |
| **Other Parts** | 12.76 | 12.77 |
| **Backward Propagation** | 24.94 | 20.96 |
| **Find Special Nodes** | - | 0.01 |
| **Calculate Derivatives** | 23.75 | 19.76 |
| **Common Nodes** | 18.09(*) | 18.09 |
| **Special Nodes** | 5.66(*) | 1.67 |
| **Data Copy** | - | 0.45 |
| **Matrix Product** | - | 1.22 |
| **Other Parts** | 1.19 | 1.20 |
| **Training Time** | 41.17 | 37.36 |

## 5 RELATED WORK

Optimization of the computation graphs has been widely studied. Some deep learning framework like TensorFlow (Abadi et al., 2016) and Theano (Bergstra et al., 2010) can generate static computation graphs and apply great optimization on them. However, these frameworks based on static computation graphs are not efficient and convenient to use when processing neural networks having dynamic computation graphs such as RNN or Tree-LSTM. In order to overcome the shortcoming, frameworks like Chainer (Tokui et al., 2015), PyTorch and DyNet (Neubig et al., 2017a) have been proposed. Unfortunately, most of them don't support the automatic operation batching and thus show a low computing performance. TensorFlow Flod (Looks et al., 2017) and DyNet (Neubig et al., 2017b) proposed their own strategies to achieve that goal and bring considerable performance improvement. We have discussed them in details in the second section. However, there are still operation batching opportunities missed by them and it's our work which makes good use of those missed ones.

## 6 CONCLUSION

In this work, we focus on the automatic batching strategies applied on dynamic computation graphs of the neural networks. The existing strategies will miss some batching opportunities in the backward propagation of the training phase. We point out that the calculation of the derivative w.r.t the result of one operation is actually a summation where the calculation of terms can be batched. We design a strategy to take advantage of those missed opportunities and implement it using DyNet toolkit. The experiments show that our implementation performs very well and brings further improvement to the training speed of different difficult-to-batch benchmarks.

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
