# OpenReview forum: "An Automatic Operation Batching Strategy for the Backward Propagation of Neural Networks Having Dynamic Computation Graphs"
_ICLR.cc/2019/Conference_

### Official Review · AnonReviewer1 · 2018-11-01
**Nice idea, but a limited study presented with lack of clarity**

**Rating:** 4
**Confidence:** 4

**Review:**

# Summary

This work describes a shortcoming in existing dynamic batching strategies, namely that they operate only on the forward pass while some operations can be batched only in the backward pass. For example, the gradient of the transition matrix in a RNN consists of the sum of partial derivatives over each time step; the terms of this sum and the summation can be batched into a single matrix multiplication. The authors implement this batching strategy in DyNet and show empirically that it can lead to decent (0-25%) speedups.

# Quality

The proposed technique comes with a trade-off which is not discussed in the paper: Delaying computations until several can be batched together can increase peak memory usage. In particular, the memory requirements of a RNN would increase from O(T) to O(2T) since each forward and backward state must now be stored. (In fact, the authors use a separately allocated contiguous block of memory that they copy the states and gradients into, which would bring this to O(3T) or O(4T) memory complexity.)

A second observation that should have been made is that the potential for speedups depends on the batch size, hidden state size, and number of time steps (or tree depth). Given a small batch size and large hidden state, the batching method effectively replaces a series of outer products with a single matrix multiplication. One would expect good speedups in this scenario. On the other hand, for a large batch size with a small hidden state, the dynamic batching strategy effectively replaces a series of inner products with a single larger inner product, which would be far less beneficial. The experiments in this work use relatively small batch sizes (64), which gives little insight about whether the proposed method would lead to speedups in a wide range of models (for example, batches of 512 are common in some NLP applications).

Some smaller comments:

* Multi-threading on a multicore architecture does not necessarily imply that operations are performed sequentially.
* Input sequences in NLP are not always sentences given as sequences of words.
* The argument that padding always leads to unnecessary computation is overly simplistic; the added control flow and branching required to perform irregular computation can often make it slower than doing regular computation plus masking (additionally, sparse kernels are often memory bandwidth bound, leading to different performance properties).
* The authors say that operations of the same "type" can be batched together, but don't specify what "type" means. I assume the type is defined by both the operation as well as the shapes of its inputs and outputs?
* No distinction is made between different ways of batching and their performance characteristics. Two matrix-vector multiplications gemv(X, y1) and gemv(X, y2) can be efficiently batched as gemm(X, [y1 y2]') which reduces the number of times X needs to be loaded into working memory. This is not the case when batching distinct inputs such as gemv(X1, y1) and gemv(X2, y2). On the other hand, gemv(X1, y1) + gemv(X2, y2) can be efficiently batched as gemv([X1 X2], [y1', y2']'), reducing the number of memory accesses in the output buffer.
* Why perform 3 runs and report the fastest speed? Why not report the range, or better yet, perform more runs and report confidence intervals.

# Clarity

The writing in this paper needs significant improvement. In terms of structure, the introduction (section 1) and background (section 2) are very repetitive. The third, fourth, fifth and sixth paragraph of the introduction are effectively repeated in full in sections 2.1, 2.2, 2.3 and 3.1 respectively. On the other hand, the inclusion of table 1 at the beginning puts the reader on the wrong foot thinking that this paper will consider NMT models, whereas the paper only deals with POS tagging and sentiment analysis.

The text contains grammatical errors ("days even weeks", "The parallel computing helps"), tautological definitions ("batching [...] means organizing the same operations of computation graphs into batches", "padding, which is to pad the input sequences"), unclear use of language ("cooperating with the existing strategies"), and typographical mistakes (multiple citations are separately parenthesized). Overall, the lack of clarity inhibits the understanding of the paper.

# Originality and significance

The central contribution of this paper is relatively straightforward in retrospect, but can certainly be beneficial for the training of some particular models. I am no expert in the literature, but the authors' claim that they are the first ones to consider this technique seems justified. The paper has no reference to code, so it is hard to judge how easy it would be for practitioners to use the suggested technique.

# Summary

Pros:

  * Useful dynamic batching trick that can lead to speedups
  * Empirical evaluation compares to two existing techniques and breaks down individual components of runtime

Cons:

  * No critical look at the disadvantages of this technique such as applicability to larger batch sizes and memory usage
  * Some questionable statements and assumptions
  * Lack of formalization and clear definitions
  * Paper reads long-drawn-out, subpar writing hurts readability

---

### Official Review · AnonReviewer3 · 2018-11-02
**Straightforward method, need more analysis.**

**Rating:** 6
**Confidence:** 5

**Review:**

This paper proposed a just-in-time optimization method of neural network calculation on dynamic computation graphs. The method focused on batching summation of gradients on the backward calculation which was performed independently in conventional toolkits, and experiments on 3 LSTM tasks showed that in several settings the proposed method improved the speed of backward computation.

The proposed method is straightforward and reasonable in terms of improving the speed of the backward computation. Authors discussed the proposed method on only the neural network toolkits with a dynamic computation strategy, but this kind of optimization can be applied to any existing toolkits even which has a non-dynamic strategy. This point looks a kind of misleading of the discussion on the paper.

The paper provided a detailed analysis of time consumption on only a success-case (Table 4). Unfortunately, Table 2 and 3 showed that the proposed method does not have a global effectiveness and suggest a necessity for a further discussion about when to use the proposed method. Since this discussion can surely be conducted by comparing analyses of success and failure-cases, authors should provide analyses of all experiments.

A conceivable weakness of the method may be the increase of memory consumption. If the toolkit plan to perform batch operations for summations of gradients, it needs to store all available gradients about each use of the corresponding variables. If the variable has a large shape and is used very frequently (e.g., variables in the softmax layer), the amount of total memory consumed by its gradient tends to be a serious problem. The non-batching strategy can mitigate this problem by discarding gradient information as soon as it is propagated back to all preceding nodes. The paper does not provide any information about memory consumption but it is important to discuss this kind of perspective.

Others:
In Table 2 and 3, please provide the ratio of speeds which are more reasonable to judge the real improvement rather than the one-zero decision (showed as up/down arrows).
In Table 4, why the time of the forward propagation slightly increased?
You should write a full list of authors of the DyNet paper that the official README provided:
https://github.com/clab/dynet/blob/master/README.md

---

### Official Review · AnonReviewer2 · 2018-11-03
**Batching for back propagation**

**Rating:** 5
**Confidence:** 3

**Review:**

Batching of similar and independent operations in a neural network computation graph is a common way to improve efficiency through computational parallelism. Optimization is often applied to the computation graph by grouping independent operations into batches that can be computed in parallel.
Existing techniques typically optimizes the feed forward computation and the backward computation follows the same grouping as the feed forward computation, which may not be optimal. In particular, the authors argue that a separate batching strategy should be applied to the back propagation computation to further improve the efficiency and showed that a recurrent network can benefit from such an optimization. The proposed solution is an automatic batching strategy that work for dynamic computation graphs.

Pros:

- The results (for both CPU and GPU) show that the proposed method improves on top of two existing batching strategies (by depth and by agenda) across three different tasks.

Cons:

- The feed forward and backward (gradient) computation can be viewed as a single computation graph. As such, would applying optimization to this graph achieve the same thing? Some clarifications/discussions will be helpful.

- It is unclear why the proposed method is better for dynamic computation graph.  The benchmark results on CPU show that the proposed method is worse than the baseline for Tree-LSTM.  Will be useful to also do a similar profiling for the cases where the proposed method did not help.

---

### Meta-Review · Area_Chair1 · 2018-12-13

**Confidence:** 4
**Recommendation:** Reject

**Metareview:**

This paper describes a new batching strategy for more efficient training of deep neural nets. The idea stems from the observation that some operations can only be batched more efficiently in the backward, suggesting that batching should be different between forward and backward. The results show that the proposed method improves upon existing batch strategies across three tasks. The reviewers find the work novel, but note that it does not properly address the trade-offs made by the technique - such as memory consumption. They also argue that the writing should be improved before acceptance at ICLR.